# Impact of Carbon Ion Radiotherapy on Inoperable Bone Sarcoma

**DOI:** 10.3390/cancers13051099

**Published:** 2021-03-04

**Authors:** Shintaro Shiba, Masahiko Okamoto, Hiroki Kiyohara, Shohei Okazaki, Takuya Kaminuma, Kei Shibuya, Isaku Kohama, Kenichi Saito, Takashi Yanagawa, Hirotaka Chikuda, Takashi Nakano, Tatsuya Ohno

**Affiliations:** 1Department of Radiation Oncology, Gunma University Graduate School of Medicine, 3-39-22, Showa-machi, Maebashi, Gunma 371-8511, Japan; okamott@gunma-u.ac.jp (M.O.); s_okazaki@gunma-u.ac.jp (S.O.); cami_taku@gunma-u.ac.jp (T.K.); shibukei@gunma-u.ac.jp (K.S.); nakano.takashi2@qst.go.jp (T.N.); tohno@gunma-u.ac.jp (T.O.); 2Gunma University Heavy Ion Medical Center, 3-39-22, Showa-machi, Maebashi, Gunma 371-8511, Japan; 3Department of Radiation Oncology, Japanese Red Cross Maebashi Hospital, 389-1, Asakura-cho, Maebashi, Gunma 371-0811, Japan; hiroki.kiyohara@maebashi.jrc.or.jp; 4Department of Orthopaedic Surgery, Gunma University Graduate School of Medicine, 3-39-22, Showa-machi, Maebashi, Gunma 371-8511, Japan; isakukohama@gunma-u.ac.jp (I.K.); tyanagaw@gunma-u.ac.jp (T.Y.); chikudah@gunma-u.ac.jp (H.C.); 5Department of Orthopaedic Surgery, National Hospital Organization Takasaki General Medical Center, 36, Takamatsu-cho, Takasaki, Gunma 370-0829, Japan; saito.kenichi.rj@mail.hosp.go.jp; 6Department of Orthopaedic Surgery, Gunma Prefectural Cancer Center, 617-1, Takahayashinishi-cho, Ota, Gunma 373-8550, Japan

**Keywords:** bone sarcoma, carbon ion radiotherapy, chordoma, chondrosarcoma, osteosarcoma, dose-volume histogram analysis

## Abstract

**Simple Summary:**

The standard treatment for bone sarcoma is surgery with or without additional chemotherapy; however, complete resection of the tumor might not be possible in patients with locally advanced lesions. Management of patients with bone sarcoma who are unsuitable for surgery is challenging. Carbon ion radiotherapy (C-ion RT) was initiated in 1994 for treating various cancers in Japan and is being considered to be an effective treatment for unresectable bone sarcoma. However, there is a limited number of reports on the clinical outcomes of C-ion RT for bone sarcoma. Here, we aimed to analyze the clinical outcomes and prognostic factors among patients with unresectable bone sarcoma who were treated with C-ion RT. We found that C-ion RT had favorable overall survival and local control with low toxicity rates compared to surgery. Therefore, our results suggest a potential role for C-ion RT in the radical treatment of inoperable bone sarcoma.

**Abstract:**

Management of patients with bone sarcoma who are unsuitable for surgery is challenging. We aimed to analyze the clinical outcomes among such patients who were treated with carbon ion radiotherapy (C-ion RT). We reviewed the medical records of the patients treated with C-ion RT between April 2011 and February 2019 and analyzed the data of 53 patients. Toxicities were classified using the National Cancer Institute’s Common Terminology Criteria for Adverse Events (Version 4.0). The median follow-up duration for all patients was 36.9 months. Histologically, 32 patients had chordoma, 9 had chondrosarcoma, 8 had osteosarcoma, 3 had undifferentiated pleomorphic sarcoma, and 1 had sclerosing epithelioid fibrosarcoma. The estimated 3-year overall survival (OS), local control (LC), and progression-free survival (PFS) rates were 79.7%, 88.6%, and 68.9%, respectively. No patients developed grade 3 or higher acute toxicities. Three patients developed both grade 3 radiation dermatitis and osteomyelitis, one developed both grade 3 radiation dermatitis and soft tissue infection, and one developed rectum-sacrum-cutaneous fistula. C-ion RT showed favorable clinical outcomes in terms of OS, LC, and PFS and low rates of toxicity in bone sarcoma patients. These results suggest a potential role for C-ion RT in the management of this population.

## 1. Introduction

The standard treatment for bone sarcoma is surgery with or without additional chemotherapy [1,2,3,4,5,6]; however, complete resection of the tumor might not be possible in patients with locally advanced lesions. X-ray radiotherapy (RT) has been considered to be one of the local treatment options for patients who have bone sarcoma that is not amenable to surgery. However, there is a limited possibility of local control of bone sarcoma with X-ray RT because of its radioresistant nature and the use of restricted RT dose to protect the surrounding normal tissues [6,7,8,9].

Carbon ion (C-ion) RT was initiated in 1994 for treating various cancers in Japan [10,11,12,13,14,15,16,17]. C-ion RT was also performed for unresectable bone sarcoma, and the treatment outcomes were significantly improved compared to X-ray RT and were comparable to surgery outcomes, although the patients were in inoperable states [15,16,17]. Moreover, functional outcomes in pelvic bone sarcoma after C-ion RT were nearly equivalent to those of surgery [18]. These favorable results were due to the biological properties of C-ion RT, including a higher relative biological effectiveness (RBE) due to the high linear energy transfer (LET) in the Bragg peak when compared to X-ray RT. Furthermore, its physical properties, such as higher dose localization ability with distal tail-off due to the Bragg peak and sharp lateral penumbra, enabled the administration of high doses [19,20,21]. Additionally, C-ion beams with high LET have a superior ability to induce cell death in radioresistant and hypoxic cells than X-rays [20,21]. However, to date, there is a limited number of reports on the clinical outcomes of patients with bone sarcoma who were treated with C-ion RT. Here, we aimed to evaluate the clinical outcomes in prospectively registered patients with bone sarcoma who were treated with C-ion RT at the Gunma University Heavy Ion Medical Center (GHMC). Additionally, we analyzed the prognostic factor of C-ion RT effectiveness for bone sarcoma treatment.

## 2. Materials and Methods 

### 2.1. Patients

We reviewed the medical records of patients with bone sarcoma who were treated with C-ion RT and registered for a prospective study at the GHMC between April 2011 and February 2019. A total of 53 consecutive patients met the following eligibility criteria for this study: (1) the presence of bone sarcoma, as confirmed by histology; (2) the absence of lymph node and distant metastasis; (3) the presence of a radiographically measurable tumor; (4) not eligible for radical surgery; and (5) performance status (PS) ≤3 by Eastern Cooperative Oncology Group classification. This retrospective study procedure was reviewed and approved by the Institutional Review Board and was used only medical records. The Institutional Review Board exempted our study from an individual informed consent, and the study was approved with an opt-out of notification regarding this analysis prior to this study. All patients signed an informed consent form before the initiation of therapy.

### 2.2. Carbon Ion Radiotherapy

Heavy ion accelerator at GHMC generated C-ion beams, and according to the depth of the tumor, the beam energy was selected from 290 MeV/u, 380 MeV/u, and 400 MeV/u. We used the XiO-N system (version 4.47; the collaborated product of Elekta AB, Stockholm, Sweden, and Mitsubishi Electric, Tokyo, Japan) for treatment planning. This system incorporates a dosing engine for ion beam RT (K2dose). We calculated the clinical radiation dose based on the physical dose multiplied by the RBE of C-ion beams and expressed it in Gy (RBE) [22]. Before C-ion RT, the patients were immobilized using tailor-made fixation cushions and thermoplastic shells to acquire treatment planning computed tomography (CT) images; respiratory-gated and 4-dimensional CT images were then acquired. 

The treatment planning CT images were merged with the magnetic resonance imaging (MRI) and/or 2-deoxy-2-[18F] fluoro-D-glucose (FDG)-positron emission tomography (PET) images to precisely delineate the gross tumor volume. The clinical target volume had a margin with an anatomical compartment of muscle or bone or at least a 5-mm around the gross tumor volume to include microscopic disease. The internal margin was assessed with reference to 4-dimensional CT images for tumor movement. The planning target volume (PTV) was defined as a summation of the clinical target volume, internal margin, and setup margin. Prescribed doses were 70.4 Gy (RBE) in 16 fractions for standard bone sarcoma without chordoma cases, 67.2 Gy (RBE) in 16 fractions for sacral chordoma cases, and 64.0 Gy (RBE) in 16 fractions for close-to-spinal-cord cases. Patients were administered C-ion RT once daily, 4 days a week (Tuesday to Friday). The planning aim was to cover the PTV with at least 95% of the prescribed dose. Dose constraints were the maximum dose (D_max_) < 60 Gy (RBE) administered to the rectum, D_max_ < 50 Gy (RBE) administered to the gastrointestinal (GI) tract, D_max_ < 50 Gy (RBE) administered to the esophagus, and D_max_ < 30 Gy (RBE) administered to the spinal cord. 

### 2.3. Evaluation during Follow-Up

Patients were followed up for 1 month after completing C-ion RT, and every 3 months until 2 years after C-ion RT, and every 4–6 months thereafter. The follow-up examinations comprised routine testing of blood cell counts and chemistry and diagnostic imaging using CT, MRI, or FDG-PET. Acute and late toxicities were graded based on the Common Terminology Criteria for Adverse Events (version 4.0) of the National Cancer Institute [23]. Acute and late toxicities were evaluated as the highest grade of toxicity that occurred within 3 months and after 3 months from the initiation of C-ion RT, respectively. 

### 2.4. Statistical Analysis

All statistical analyses were performed using the Statistical Package for the Social Sciences software, version 25.0 (IBM Inc., Armonk, NY, USA). Survival was measured from the date of C-ion RT initiation to the date of death or the most recent follow-up. Local control (LC) was defined as no evidence of recurrence in the field (excluding the edge of the irradiated field) as detected by any increase in tumor size on CT and MRI or no increase in FDG uptake on FDG-PET. Progression-free survival (PFS) was measured from the date of initiation of C-ion RT to the date of either an observation of tumor progression or death from any cause. Toxicities were evaluated from the date of initiation of C-ion RT to the date of either receiving secondary treatment for tumor recurrence or death. Clinical outcomes were analyzed in all patients, and those with chordoma, and other bone sarcomas (non-chordoma), separately. Probabilities of overall survival (OS), LC, and PFS rates were calculated using the Kaplan-Meier method. We analyzed clinical outcomes separately for all patients, chordoma patients, and non-chordoma patients because the malignancy grade of chordoma and non-chordoma is different.

Additionally, we demonstrated dose-volume histogram (DVH) analysis to evaluate the prognostic factor of C-ion RT. We assessed the minimum dose that covered 95% and 98% of the target volume (D_95_ and D_98_) for the gross tumor volume (GTV), the percentage of the GTV that received at least 60 and 64 Gy (RBE) (V_60_ and V_64_), and the GTV that received less than 60 and 64 Gy (RBE) (V_<60_ and V_<64_) based on the DVH. Next, we evaluated whether histology, sex, age, presence or absence of chemotherapy, PS, presence or absence of prior treatment (surgery and/or chemotherapy), minimum distance of the tumor from gastrointestinal (GI) tract, GTV, GTV D_95_, GTV D_98_, GTV V_60_, GTV V_64_, GTV V_<60_, and GTV V_<64_ in OS and LC could be prognostic factors using the log-rank test. We determined the strength of associations between a minimum distance of the tumor from the GI tract, GTV D_95_, GTV D_98_, GTV V_60_, GTV V_64_, GTV V_<60_, and GTV V_<64_ using the Pearson correlation coefficient. Statistical significance was defined as a *p*-value of <0.05.

## 3. Results 

### 3.1. Patient Characteristics

The clinical characteristics of the 53 eligible patients are summarized in Table 1. The median follow-up duration after the initiation of C-ion RT for all patients was 36.9 months (range: 4.4–96.4), and that for all surviving patients was 42.3 months (range: 9.4–96.4). Histologically, 32 patients had chordoma, 9 patients had chondrosarcoma, 8 patients had osteosarcoma, 3 patients had undifferentiated pleomorphic sarcoma, and 1 patient had sclerosing epithelioid fibrosarcoma. Thirteen of the 53 patients underwent prior treatment with C-ion RT and the other 40 patients were treatment naïve. In the prior treatment of C-ion RT, six osteosarcoma patients were administered chemotherapy, five chordoma patients underwent surgery, and each patient with chondrosarcoma and sclerosing epithelioid fibrosarcoma underwent surgery. No patient has received C-ion RT as prophylactic post-operative irradiation. Seventeen patients had bladder and bowel dysfunction before C-ion RT. The median distance between the tumor and the nearest intestinal tract was 3.4 mm (range: 0–96).

### 3.2. Clinical Outcomes in All Patients 

Figure 1 shows the MRI and FDG-PET scans in a patient with sacral chondrosarcoma before and after C-ion RT and dose distribution with C-ion RT. 

Figure 2 shows the curves of OS, LC, and PFS in all patients. The estimated 3-year OS, LC, and PFS rates in all patients were 79.7%, 88.6%, and 68.9%, respectively, and the 5-year OS, LC, and PFS rates were 79.7%, 73.8%, and 48.6%, respectively. Six of 17 patients with bladder and bowel dysfunction showed improvement of function, and the remaining 11 patients did not show improvement. 

There were significant differences in OS and PFS by the histological type (chordoma or non-chordoma) (*p* < 0.01 and *p* < 0.05) while the difference in LC did not reach the significant level (*p* = 0.057). There were significant differences in clinical outcomes between chordoma and non-chordoma patients; therefore, we analyzed prognostic factors for these two histological types separately. 

In the analysis of correlations between the minimum distance of the tumor from the GI tract and each of the DVH parameters, negative correlations were observed for GTV V_<60_ and GTV V_<64_ (R = −0.6 and −0.5, respectively). In contrast, there were no significant correlations between the minimum distance of the tumor from the GI tract and either of the GTV D_95_, GTV D_98_, GTV V_60_, and GTV V_64_.

### 3.3. Clinical Outcomes in Chordoma Patients

Thirty-two of 53 patients had chordoma. These patients were analyzed, and patient characteristics are summarized in Table 1. The median follow-up duration after the initiation of C-ion RT in all patients with chordoma was 40.2 months (range: 9.4–96.4), and in all surviving patients with chordoma, it was 41.7 months (range: 9.4–96.4). The estimated 3-year OS, LC, and PFS rates were 91.3%, 92.5%, and 79.5%, respectively, and 5-year OS, LC, and PFS rates were 91.3%, 84.8%, and 57.8%, respectively (Figure 2). 

In the prognostic factor analysis, PS (PS = 0–1, or PS = 2–3) was a significant prognostic factor for OS. GTV (GTV ≤ 300 cm^3^, or GTV > 300 cm^3^) and GTV V_<60_ (GTV V_<60_ ≤ 1 cm^3^, or GTV V_<60_ > 1 cm^3^) were significant prognostic factors for LC (both *p* < 0.05). The results of the analysis for the other factors are shown in Table 2.

At the time of the analysis, three patients had local recurrence, and three other patients had the edge of irradiated field recurrence. All patients with the edge of irradiated field recurrence also developed lung metastases. Total 5 of 32 patients had distant metastases (4 lung metastases and 1 bone metastases). Four of all patients with recurrence and metastases underwent C-ion RT as a salvage treatment.

### 3.4. Clinical Outcomes in Non-Chordoma Patients

Overall, 21 of 53 patients had non-chordoma. These patients were analyzed, and patient characteristics are summarized in Table 1. The median follow-up duration after the initiation of C-ion RT in all patients with non-chordoma was 24.8 months (range: 4.4–85.7), and that in all surviving patients with non-chordoma was 47.3 months (range: 9.5–95.7). The estimated 3-year OS, LC, and PFS rates were 60.7%, 82.2%, and 52.3%, respectively, and the 5-year OS, LC, and PFS rates were 60.7%, 54.8%, and 35.4%, respectively (Figure 2). Pathologically, 3-year OS and LC in osteosarcoma were 36.5% and 87.5%, respectively, and the corresponding ones in chondrosarcoma were 59.3% and 60%, respectively (*p* = 0.316 and *p* = 0.376).

In the prognostic factor analysis, there were no factors associated with prognosis in LC and OS (Table 2). However, the use of chemotherapy before or after C-ion RT significantly improved OS in osteosarcoma; 3-year OS in the presence and absence of chemotherapy were 50% and 0%, respectively (*p* < 0.01).

At the time of the analysis, five patients had local recurrence, and one patient had the edge of irradiated field recurrence. Three of five patients with local recurrence also developed distant metastases. A total 6 of 21 patients had distant metastases (4 lung metastases, 1 bone metastases, and 1 subcutaneous metastasis). Five of all patients with recurrence and metastases underwent C-ion RT as a salvage treatment.

### 3.5. Toxicities

All the observed acute and late toxicities are listed in Table 3. No patients developed grade 3 or higher acute toxicities. Regarding the late toxicities, five patients developed grade 3 toxicities. Three of five patients developed both grade 3 radiation dermatitis and osteomyelitis and required treatment of osteomyelitis curettage, gluteus muscle flap, and intravenous antibiotics. One of these three patients developed local recurrence and underwent salvage treatments with re-irradiation C-ion RT and surgery. After salvage treatments, this patient developed osteomyelitis with difficulty to control. Another patient developed both grade 3 radiation dermatitis and soft tissue infection and required drainage and intravenous antibiotics. Furthermore, the last one of these patients developed a rectum-sacrum-cutaneous fistula and required surgery of colostomy, sacral rectal resection, and rectus abdominis flaps [24]. A total of 6 patients developed grade 2 bone fracture and 11 patients developed grade 2 peripheral neuropathy. No patients developed newly occurred bladder and bowel dysfunction.

## 4. Discussion 

In this study, we evaluated clinical outcomes in patients with bone sarcoma who were treated with C-ion RT. The 3-year OS, LC, and PFS rates were 79.7%, 88.6%, and 68.9%, respectively, the 5-year OS, LC, and PFS rates were 79.7%, 73.8%, and 48.6%, respectively, and five patients (9.4%) developed grade 3 toxicity. We observed favorable efficacy and a low rate of toxicity when C-ion RT was used to manage bone sarcoma; however, only one patient developed osteomyelitis with difficulty to control after the salvage treatment with re-irradiation of C-ion RT and surgery for local recurrence, suggesting that performing salvage treatment may be associated with a risk of complication. This study showed comparable clinical outcomes in efficacy and safety to those presented in the previous studies on C-ion RT for bone sarcoma that used similar therapeutic schedules of dose fractionation [15,16,17]. In previous reports on bone sarcoma, 5-year OS and LC in X-ray RT were 50–70% and 27–67%, respectively, and those in proton beam therapy were 67–84% and 62–81%, respectively [7,8,9,25,26,27]. Therefore, based on the clinical outcomes, C-ion RT used in this study was comparable to proton beam therapy but superior to X-ray RT. Furthermore, these results were comparable to those of surgery despite having included patients who were unsuitable for resection [6,7,28,29,30,31,32], suggesting that C-ion RT could be a treatment option for such patients. 

We demonstrated prognostic factors that were important for LC and OS. According to our results, better PS was associated with higher OS in patients with chordoma, while small GTV and low GTV V_<60_ were associated with higher LC in these, and the use of chemotherapy (neoadjuvant and/or adjuvant) contributed to the higher OS in patients with osteosarcoma. In DVH analysis, we focused on the volume that was not irradiated to a dose of a particular level (V_<60_ and V_<64_) in tumor control, as the result of a previous study on dose escalation in C-ion RT for bone and soft tissue sarcomas showed a significant improvement in local control between under 57.6 Gy (RBE) and over 64 Gy (RBE) dose administration [33]. Furthermore, we have considered that the tumor located close-to-spinal-cord could be controlled by 64 Gy (RBE). Therefore, we have analyzed 60 and 64 Gy (RBE) in DVH analysis. We focused on V_<X_ instead of D_X_ and V_X_ because the parameters of D_X_ and V_X_ were calculated using a percentage of the tumor volume, even if this parameter was high, it did not necessarily mean that the unirradiated tumor volume is small in the larger tumor. Therefore, larger tumors are considered to have a higher risk of recurrence due to the higher amount of remaining tumor cells left in the unirradiated area even if D_X_ and V_X_ were high. Furthermore, the DVH parameters of V_<X_ were synonymous with positive margins in surgery, and it is considered that the risk of local recurrence was high. In our study, in patients who were administered lower dose, it was due to the priority given to the dose constraint of the GI tract or skin. However, the parameter V_<60_ was not a prognostic factor for LC in non-chordoma patients. The reasons for these results in the non-chordoma patients could be the few events of local failure and the low number of the patient cohorts. Further analysis would be needed to reveal the prognostic factor of C-ion RT using many patients with non-chordoma and a long follow-up period. 

Administration of a sufficient dose is important for LC improvement, and it was reflected as V_<60_ in chordoma patients. There was a negative correlation between the GTV V_<60_ and the distance of the tumor from the GI tract. To shorten the distance between the tumor and the GI tract, a spacer could be used [34,35]. A spacer can be placed before the C-ion RT using Gore-Tex sheets (W.L. Gore and Associates, Newark, DE, USA), which would physically separate the tumor from the GI tract and enable administration of a sufficient dose of C-ion RT to control the sarcoma. Another option is using a bioabsorbable polyglycolic acid (PGA) spacer, which has been recently developed [36]. Hitherto, there was a risk of infection due to spacer placement using Gore-Tex sheets for a long time after C-ion RT. The PGA spacer might enable safer conditions than Gore-Tex sheets because the PGA spacer will be absorbed after C-ion RT, and the risk of infection will be reduced. Therefore, a spacer placement is an additional option to improve LC in C-ion RT in the case of a short distance between the tumor and the GI tract.

In the cases of chordoma, the 3-year OS and LC rates were 91.3% and 92.5%, respectively, and toxicities were tolerable with no bladder and bowel dysfunction caused by C-ion RT. Additionally, none of the patients who had bladder and bowel dysfunction before C-ion RT had worsening symptoms after this procedure. These results were comparable to those in surgery. Previously, Nishida et al. compared the results of surgery with those of C-ion RT in sacral chordoma and concluded that the oncologic results in both treatments were comparable [32]. It is necessary to consider whether surgery or C-ion RT should be recommended for each patient on an individual basis. Our study found that even a small volume could cause local recurrence if the dose for GTV is not sufficient (i.e., GTV V_<60_). In these patients, surgery might be recommended. In contrast, C-ion RT might be recommended for patients with sacral chordoma located in the second sacral spine or higher because the risk of bladder-bowel dysfunction is increased. If the patient has chordoma located in the second sacral spine or higher with close proximity to the GI tract, the patient should receive surgical spacer placement, and the subsequent C-ion RT might overcome these toxicities and restrictions [34,35]. 

C-ion RT for non-chordoma patients improved clinical outcomes in this study, compared to those in previous reports on X-ray RT [6,8,9]; however, no prognostic factors were identified except for the use of chemotherapy for osteosarcoma in OS. This result suggested that neoadjuvant and/or adjuvant chemotherapy improved OS in osteosarcoma, and we have considered a possibility that concurrent use of C-ion RT with chemotherapy might further improve OS because it can prevent micrometastases. Although C-ion RT for bone sarcoma is performed without concurrent chemotherapy, other cancers are being treated by C-ion RT with concurrent chemotherapy achieving good clinical results in terms of OS and LC [10]. Concurrent use of chemotherapy and C-ion RT for sarcomas, especially for aggressive types such as osteosarcoma and mesenchymal chondrosarcoma, might be beneficial if it can provide similar effects as it did in other cancers. 

Our study had some limitations. First, various types of bone sarcomas were analyzed without using histological distinctions in this study. It would be necessary to analyze histological-specific clinical data of C-ion RT efficacy and safety. Further analysis with many patients with bone sarcoma is warranted. Second, although this study showed favorable clinical outcomes, the follow-up duration was insufficient to evaluate the long-term efficacy of C-ion RT. However, the follow-up duration in this study was sufficient to confirm the safety of C-ion RT because long-term radiation-related adverse events are uncommon except for radiation-induced malignancies.

## 5. Conclusions 

C-ion RT resulted in favorable clinical outcomes in terms of OS, LC, and PFS and low rates of toxicity in patients with bone sarcoma. These results suggested a potential role for C-ion RT in the radical treatment of patients with bone sarcoma who are unsuitable for surgery. Additionally, we found that tumor volume and DVH analysis of GTV V_<60_ might be prognostic factors for LC in patients with chordoma. To date, the actual GTV volume that was not sufficiently irradiated has not been analyzed in other cancers, and it may be worth examining it in other cancers treated with C-ion RT as well.

## Figures and Tables

**Figure 1 cancers-13-01099-f001:**
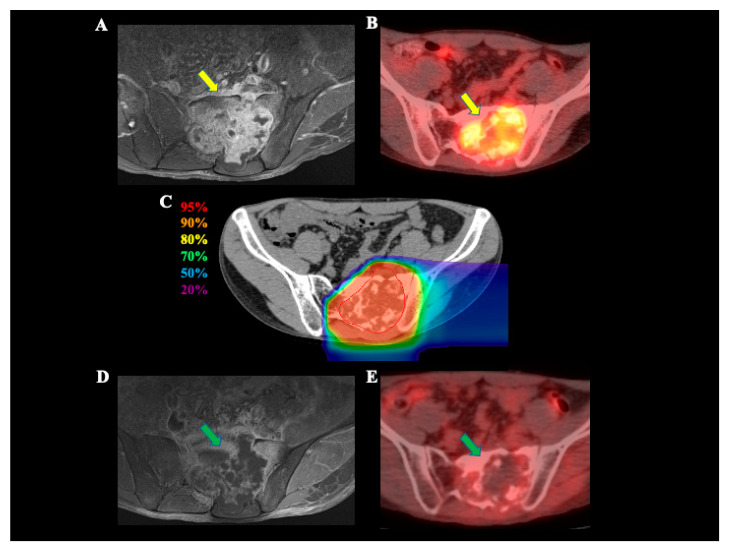
A 24-year-old male with sacral chondrosarcoma treated with C-ion RT. (**A**) Contrast-enhanced MRI before treatment. The yellow arrow shows the tumor with contrast enhancement. (**B**) FDG-PET before treatment. The yellow arrow shows the tumor with abnormal FDG uptake. (**C**) Dose distribution on axial CT images. The area within the red outline is GTV. Highlighted are the 95% (red), 90% (orange), 80% (yellow), 70% (green), 50% (blue), and 20% (purple) isodose curves (100% was 70.4 Gy [RBE]). (**D**) Contrast-enhanced MRI 3 months after treatment. The tumor contrast effect is decreased compared to that before treatment (green arrow). (**E**) FDG-PET 3 months after treatment. FDG uptake is decreased compared to that before treatment (green arrow). *Abbreviations:* C-ion RT, carbon ion radiotherapy; CT, computed tomography; FDG-PET, 2-deoxy-2-[18F] fluoro-D-glucose (FDG)-positron emission tomography (PET); GTV, gross tumor volume; MRI, magnetic resonance imaging; RBE, relative biological effectiveness.

**Figure 2 cancers-13-01099-f002:**
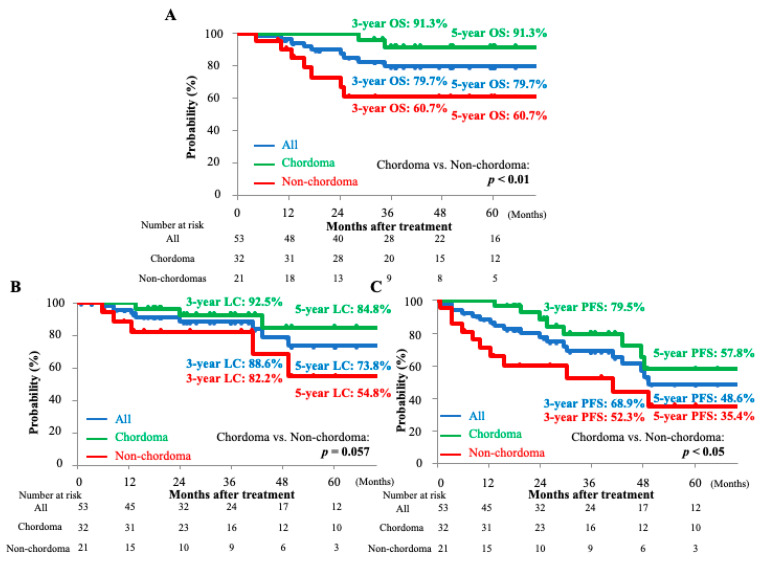
Kaplan-Meier curves in all (blue line), chordoma (green line), and non-chordoma (red line) patients. The number at risk is shown below the figure. (**A**) Overall survival. (**B**) Local control. (**C**) Progression-free survival. Abbreviations: LC, local control; OS, overall survival; PFS, Progression-free survival, vs., versus.

**Table 1 cancers-13-01099-t001:** Patient characteristics.

Characteristics	All	Chordoma	Non-Chordoma
Number	53	32	21
Age, years, median (range)	67 (14–84)	67 (27–84)	54 (14–83)
Sex, number (%)			
Male	32 (60.4%)	24 (75%)	8 (38.1%)
Female	21 (39.6%)	8 (25%)	13 (61.9%)
Performance status			
0–1	47 (88.7%)	30 (93.7%)	17 (81.0%)
2–3	6 (11.3%)	2 (6.3%)	4 (19.0%)
Histology, number (%)			
Chordoma	32 (60.4%)	32 (100%)	None
Chondrosarcoma	9 (17.0%)	None	9 (42.9%)
Osteosarcoma	8 (15.1%)	None	8 (38.1%)
Undifferentiated pleomorphic sarcoma	3 (5.6%)	None	3 (14.3%)
Sclerosing epithelioid fibrosarcoma	1 (1.9%)	None	1 (4.7%)
Tumor site, number (%)			
Pelvis	49 (92.5%)	30 (93.7%)	19 (90.6%)
Shoulder	1 (1.9%)	None	1 (4.7%)
Spine	3 (5.6%)	2 (6.3%)	1 (4.7%)
Prior treatment of C-ion RT			
Chemotherapy	6 (11.3%)	None	6 (28.6%)
Surgery	7 (13.2%)	5 (15.6%)	2 (9.5%)
Treatment naïve	40 (75.5%)	27 (84.4%)	13 (61.9%)
Tumor size, mm, median (range)	95 (15–175)	87.5 (15–175)	100 (16–160)
Minimum distance of tumor from GI, mm, median	3.4 (0.1–96.0)	2.3 (0.1–67.5)	4.0 (0.1–96.0)
GTV volume, cm^3^, median (range)	215.6 (1.6–2074.3)	205.7 (1.6–2074.3)	225.4 (2.2–1551.2)
Dose of C-ion RT, Gy (RBE)			
64.0	3 (5.6%)	2 (6.3%)	1 (4.7%)
67.2	30 (56.6%)	30 (93.7%)	None
70.4	20 (37.8%)	None	20 (95.3)
GTV D_98_, Gy (RBE), median (range)	66.15 (43.77–70.60)	64.24 (43.77–66.80)	68.94 (62.77–70.60)
GTV D_95_, Gy (RBE), median (range)	66.73 (58.36–70.70)	66.48 (58.36–64.27)	69.36 (63.54–70.70)
GTV V_64_, %, median (range)	99.5 (47.1–100)	98.3 (47.1–100)	100 (53.7–100)
GTV V_60_, %, median (range)	99.9 (82.4–100)	99.7 (82.4–100)	100 (98.7–100)
GTV V_<64_, cm^3^, median (range)	0.9 (0–144.0)	2.5 (0–144.0)	0 (0–62.9)
GTV V_<60_, cm^3^, median (range)	0.1 (0–32.3)	0.6 (0–32.3)	0 (0–19.4)

Abbreviations: C-ion RT, carbon ion radiotherapy; D_95_ and D_98_, the minimum dose that covered 95% and 98% of the target volume; GTV, gross tumor volume; RBE, relative biological effectiveness; V_60_ and V_64_, percentage of the GTV volume that received at least 60 and 64 Gy (RBE); V_<60_ and V_<65_, the target volume that received less than 60 and 65 Gy (RBE).

**Table 2 cancers-13-01099-t002:** Univariate analysis of overall survival and local control.

Clinical Factor	Chordoma (*n* = 32)					Clinical Factor	Non-Chordoma (*n* = 21)				
	*n*	3-y OS (%)	*p*	3-y LC (%)	*p*		*n*	3-y OS (%)	*p*	3-y LC (%)	*p*
Age						Age					
≤67	16	100	0.059	100	0.611	≤54	11	87.5	0.056	81.8	0.638
>67	16	83.9		84.8		>54	10	33.8		85.7	
Sex						Sex					
Male	24	93.8	0.816	100	0.061	Male	8	43.8	0.259	72.9	0.595
Female	8	83.3		71.4		Female	13	73.4		88.9	
Chemotherapy						Chemotherapy					
Presence	0	NA	NA	NA	NA	Presence	14	61.1	0.785	90.9	0.131
Absence	32	91.3		92.5		Absence	7	60.0		66.7	
Performance status						Performance status					
0–1	30	95.5	0.042 *	92.0	0.649	0–1	17	56.8	0.510	83.9	0.992
2–3	2	50.0		100		2–3	4	75.0		75.0	
Prior treatment						Prior treatment					
Presence	5	89.9	0.554	90.9	0.435	Presence	8	59.8	0.713	90.0	0.422
Absence	27	100		100		Absence	13	60.0		71.4	
Distance of tumor-GI						Distance of tumor-GI					
≤3 mm	18	85.6	0.574	86.9	0.134	≤3 mm	14	20.8	0.070	75.0	0.849
>3 mm	14	100		100		>3 mm	7	76.6		84.6	
Distance of tumor-GI						Distance of tumor-GI					
≤5 mm	22	87.4	0.718	88.9	0.256	≤5 mm	11	51.4	0.908	87.5	0.671
>5 mm	10	100		100		>5 mm	10	67.5		77.8	
GTV volume, cm^3^						GTV volume, cm^3^					
≤300	21	100	0.067	100	0.014 *	≤300	14	64.7	0.418	77.9	0.165
>300	11	77.1		77.9		>300	7	51.4		100	
GTV D_98_, Gy (RBE)						GTV D_98_, Gy (RBE)					
≤64.0	14	77.1	0.269	92.3	0.969	≤69.0	11	60.0	0.58	88.9	0.674
>64.0	18	100		93.3		>69.0	10	60.0		77.8	
GTV D_95_, Gy (RBE)						GTV D_95_, Gy (RBE)					
≤66.0	15	77.1	0.269	92.9	0.895	≤69.4	11	78.8	0.33	87.5	0.364
>66.0	17	100		92.9		>69.4	10	40.0		76.2	
GTV V_64_, %						GTV V_64_, %					
≤98	13	74.1	0.216	91.7	0.867	≤98	1	100	0.821	100	0.520
>98	19	100		93.8		>98	20	58.2		81.1	
GTV V_60_, %						GTV V_60_, %					
≤98	3	100	0.372	100	0.553	≤98	0	NA	NA	NA	NA
>98	29	89.9		91.5		>98	21	60.7		82.2	
GTV V_<64_, cm^3^						GTV V_<65_, cm^3^					
≤1	12	100	0.508	100	0.13	≤1 cm^3^	16	63.3	0.324	79.4	0.854
>1	20	84.4		86.6		>1 cm^3^	5	50		100	
GTV V_<60_, cm^3^						GTV V_<60_, cm^3^					
≤1	20	92.3	0.876	100	0.035 *	≤1	18	66.1	0.26	81.1	0.42
>1	12	90		81.5		>1	3	33.3		100	

Abbreviations: D_95_ and D_98_, the minimum dose that covered 95% and 98% of the target volume; GTV, gross tumor volume; LC, local control; OS, overall survival; RBE, relative biological effectiveness; V_60_ and V_64_, percentage of the GTV volume that received at least 60 and 64 Gy (RBE); V_<60_ and V_<64_, the target volume that received less than 60 and 64 Gy (RBE). * *p* < 0.05.

**Table 3 cancers-13-01099-t003:** Acute and late toxicities according to CTCAE, version 4.0 (*n* = 53).

**Acute Toxicities**					
	**Grade 0**	**Grade 1**	**Grade 2**	**Grade 3**	**Grade 4**
Dermatitis	26	21	6	0	0
GI tract	51	2	0	0	0
Urinary	53	0	0	0	0
Neuropathy	46	6	1	0	0
Infection	53	0	0	0	0
**Late Toxicities**					
	**Grade 0**	**Grade 1**	**Grade 2**	**Grade 3**	**Grade 4**
Dermatitis	28	17	3	5	0
GI tract	50	1	1	1	0
Urinary	47	3	3	0	0
Bone fracture	45	4	4	0	0
Neuropathy	17	24	12	0	0
Infection	48	0	0	5	0

Abbreviations: CTCAE, Common Terminology Criteria for Adverse Events; GI, gastrointestinal tract.

## Data Availability

The datasets generated for this study are available on request to the corresponding author.

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
