# Peer review of "Impact of Carbon Ion Radiotherapy on Inoperable Bone Sarcoma"

_cancers, 2021, doi:10.3390/cancers13051099_

Round 1

Reviewer 1 Report

The results of a prospective registry of 53 patients with unresectable bone tumors who received carbon ion irradiation are shown. The median follow-up time is 36.9 months. Histologies were mainly chordoma (32), chondrosarcoma (9), osteosarcoma (8), undifferentiated pleomorphic sarcoma (5), and sclerosing epithelioid fibrosarcoma (1). Very high local control rates (88.6% at 3 years) were described with 92.5% for chordomas and 82.2% for non-chordomas. Acute toxicities were mild, whereas grade 3 late toxicities (dermatitis, GI tract, infection) were reported in 5 patients. These results support the use of carbon ion radiotherapy for unresectable bone sarcomas. However, a detailed comparison to proton irradiation and standard radiography is lacking.

 Mayor's comment:

- The results on outcome after carbon ion irradiation of patients with unresectable bone sarcoma should be discussed in detail with the previously published results of patients treated with either proton irradiation or standard x-ray.

Minor Comments:

- Follow-up schedule: It is stated that imaging was done every 3 months. For how long was this done and when were the intervals extended?

- Characteristics: Chondrosarcoma. Does this rule out dedifferentiated chondrosarcoma or myxoid chondrosarcoma? Has grading been performed?

- Tumor size: does the median tumor size of 95 mm include patients with prior surgery? Was tumor size measured for these patients before or after surgery?

Author Response

The results of a prospective registry of 53 patients with unresectable bone tumors who received carbon ion irradiation are shown. The median follow-up time is 36.9 months. Histologies were mainly chordoma (32), chondrosarcoma (9), osteosarcoma (8), undifferentiated pleomorphic sarcoma (5), and sclerosing epithelioid fibrosarcoma (1). Very high local control rates (88.6% at 3 years) were described with 92.5% for chordomas and 82.2% for non-chordomas. Acute toxicities were mild, whereas grade 3 late toxicities (dermatitis, GI tract, infection) were reported in 5 patients. These results support the use of carbon ion radiotherapy for unresectable bone sarcomas. However, a detailed comparison to proton irradiation and standard radiography is lacking.

Response: We sincerely thank you for evaluating our manuscript and for the encouraging comments. After revising our manuscript according to your suggestions, we have resubmitted it for further review. We hope we have addressed all your concerns.

Major comment:

The results on outcome after carbon ion irradiation of patients with unresectable bone sarcoma should be discussed in detail with the previously published results of patients treated with either proton irradiation or standard x-ray.

Response: We thank you for the important comment. We added the following sentence in the first paragraph of the “Discussion” (line 274–278) to compare photon/proton therapy to carbon ion radiotherapy (C-ion RT). Additionally, we added scholarly references for proton beam therapy. “In previous reports on bone sarcoma, 5-year OS and LC in X-ray RT were 50–70% and 27–67%, respectively, and those in proton beam therapy were 67–84% and 62–81%, respectively. Therefore, based on the clinical outcomes, C-ion RT used in this study was comparable to proton beam therapy but superior to X-ray RT.”

Minor Comments:

Follow-up schedule: It is stated that imaging was done every 3 months. For how long was this done and when were the intervals extended?

Response: We apologize for the insufficient details about the duration of imaging. Patients were followed up for 1 month after completing C-ion RT, and every 3 months until 2 years after C-ion RT, and every 4–6 months thereafter. We included the sentence in the section “Evaluation during follow-up” under “Materials and Methods” (line 115–116).

Characteristics: Chondrosarcoma. Does this rule out dedifferentiated chondrosarcoma or myxoid chondrosarcoma? Has grading been performed?

Response: We thank you for the important comment. In chordoma patients, 1 of 9 patients had dedifferentiated chondrosarcoma. The other 8 patients had conventional chondrosarcoma in which 3 had grade 1 chondrosarcoma, 3 grade 2 chondrosarcoma, and 2 were not graded.

Tumor size: does the median tumor size of 95 mm include patients with prior surgery? Was tumor size measured for these patients before or after surgery?

Response: We thank you for the important comment. The patients who received prior surgery were included in the patient cohort with a median tumor size of 95 mm. For the patients who received surgery before C-ion RT, the tumor size was measured after surgery. In our cohort, no patient received C-ion RT as adjuvant or prophylactic therapy.

Reviewer 2 Report

In this manuscript, Shiba et al. reported the clinical outcomes among patients with inoperable bone sarcoma, who were treated with carbon ion radiotherapy (C-ion RT). They reviewed the medical records of 53 patients and found that C-ion RT showed favorable clinical outcomes in terms of OS, LC, and PFS and low rates of toxicity in bone sarcoma patients, suggesting that C-ion RT can be an option for the management of patients with inoperable bone sarcoma.

Comments:

  1. The nature of the study and the ethical statement is confusing. The study subjects were registered for prospective study at GHMC between April 2011 and February 2019. However, the study was approved by Institutional Review Board (IRB) on October 11, 2019, after the patients have registered for this study. Were these patients recruited without IRB approval? If the approved IRB only covers reviewing medical records, did the investigators obtain the informed consent to release their medical records? The authors only mentioned that they obtained the informed consent form before the initiation of therapy. It is unclear if the consent form includes releasing medical records for research.
  2. The authors cited several studies that compare the clinical outcomes of C-ion RT, surgery, and X-ray RT for patients in inoperable states (line 63-67). This study seems to reach the same conclusions as other previously published studies. Although it is important to keep up to date the clinical outcomes of C-ion RT treatment, it is unclear how this study contributes to the field and advances the treatment for patients with inoperable bone sarcoma.
  3. The analysis of the prognostic factors of C-ion RT effectiveness for bone sarcoma treatment is problematic. The authors divided 53 patients into two sub-groups: chordoma group (N=32) and non-chordoma (n=21), and then further separated each sub-groups into groups with an even smaller sample size. Some of such groups only have one or two subjects. For example, Performance status was identified as one significant prognostic factor for OS; however, only two subjects were in one of the groups. The results obtained from the statistical analysis using such a small sample size are not convincing. Have the authors tried any multivariate analysis?
  4.  The study did not directly compare C-ion RT with other treatment options, such as X-ray RT. The study would be more interesting if a group of patients treated with X-ray RT was included.

Author Response

In this manuscript, Shiba et al. reported the clinical outcomes among patients with inoperable bone sarcoma, who were treated with carbon ion radiotherapy (C-ion RT). They reviewed the medical records of 53 patients and found that C-ion RT showed favorable clinical outcomes in terms of OS, LC, and PFS and low rates of toxicity in bone sarcoma patients, suggesting that C-ion RT can be an option for the management of patients with inoperable bone sarcoma.

Response: We sincerely thank you for evaluating our manuscript and for the encouraging comments.

The nature of the study and the ethical statement is confusing. The study subjects were registered for prospective study at GHMC between April 2011 and February 2019. However, the study was approved by Institutional Review Board (IRB) on October 11, 2019, after the patients have registered for this study. Were these patients recruited without IRB approval? If the approved IRB only covers reviewing medical records, did the investigators obtain the informed consent to release their medical records? The authors only mentioned that they obtained the informed consent form before the initiation of therapy. It is unclear if the consent form includes releasing medical records for research.

Response: We thank you for the important comment. This is a retrospective study, and we used only medical records. IRB exempted our study from an individual informed consent, and the study was approved with an opt-out of notification regarding this analysis prior to this study.

The authors cited several studies that compare the clinical outcomes of C-ion RT, surgery, and X-ray RT for patients in inoperable states (line 63-67). This study seems to reach the same conclusions as other previously published studies. Although it is important to keep up to date the clinical outcomes of C-ion RT treatment, it is unclear how this study contributes to the field and advances the treatment for patients with inoperable bone sarcoma.

Response: We thank you for the comment. The number of reports on C-ion RT for bone sarcoma and the number of facilities that have reported the clinical outcomes of C-ion RT for bone sarcoma are limited. Therefore, we considered it important to show the reproducibility of the treatment results of previous reports in other facilities. This report suggests that comparable results can be obtained using the same fractionation schedules.

The analysis of the prognostic factors of C-ion RT effectiveness for bone sarcoma treatment is problematic. The authors divided 53 patients into two sub-groups: chordoma group (N=32) and non-chordoma (n=21), and then further separated each sub-groups into groups with an even smaller sample size. Some of such groups only have one or two subjects. For example, Performance status was identified as one significant prognostic factor for OS; however, only two subjects were in one of the groups. The results obtained from the statistical analysis using such a small sample size are not convincing. Have the authors tried any multivariate analysis?

Response: We thank you for the insightful comment. We agree with you that the statistical analysis using such a small sample size is not convincing. We used multiple logistic regression for multivariate analysis. We used the log-rank test to identify potential prognostic factors (P < 0.10) for the overall survival (OS) of chordoma and non-chordoma (analyzed by age, performance status, and GTV volume in OS of chordoma, and age and distance of tumor-GI [≤3 mm or >3 mm] in OS of non-chordoma, respectively), and local control (LC) of chordoma (analyzed by sex, GTV volume, and GTV V<60). In LC of non-chordoma analysis, no factors had P < 0.10 in the log-rank test; therefore, multivariate analysis was performed using the two factors with low P values in the log-rank test (analyzed by GTV volume and presence or absence of chemotherapy). However, there were no significant prognostic factors for OS and LC of chordoma and non-chordoma patients in multivariate analysis. However, the analysis was limited by the small number of patients and events (death and local failure).

The study did not directly compare C-ion RT with other treatment options, such as X-ray RT. The study would be more interesting if a group of patients treated with X-ray RT was included.

Response: We thank you for the insightful comment. We agree that the direct comparison of C-ion RT with X-ray RT would be interesting. However, data about clinical outcomes in radical X-ray RT for bone sarcoma in our institution is limited. Therefore, direct comparison of X-ray RT and C-ion RT is difficult and could not be performed in this analysis.

Reviewer 3 Report

In this manuscript the authors deal with their results in the treatment of inoperable bone sarcoma by carbon ion radiotherapy.

The manuscript is well written and organized.

I would add some comments and suggestions:

  1. The authors should report not only the 3-years but also the 5-year OS, LC, and PFS.
  2. It would be desirable that a group-match study be performed to verify the real effectiveness of C-ion Rt versus RT. This comparison would help to understand the effectiveness of C-ion RT.
  3. Authors should explain why they have decided to divide the entire population into two groups based on histology.

I there any difference in terms of OS, LC, and PFS between the group of patients who received previous treatment (chemo, surgery) versus the naïve group of patients who received C-ion RT?

Author Response

In this manuscript the authors deal with their results in the treatment of inoperable bone sarcoma by carbon ion radiotherapy.

The manuscript is well written and organized.

I would add some comments and suggestions:

Response: We sincerely thank you for evaluating our manuscript and for the encouraging comments.

The authors should report not only the 3-years but also the 5-year OS, LC, and PFS.

Response: We thank you for the important comment. We added the 5-year OS, LC, and PFS in the “Results” section and “Figure 2”. The following sentences “The estimated 3-year OS, LC, and PFS rates in all patients were 79.7%, 88.6%, and 68.9%, respectively, and the 5-year OS, LC, and PFS rates were 79.7%, 73.8%, and 48.6%, respectively.” and “The estimated 3-year OS, LC, and PFS rates were 60.7%, 82.2%, and 52.3%, respectively, and the 5-year OS, LC, and PFS rates were 60.7%, 54.8%, and 35.4%, respectively (Figure 2).” were added to the “Results “section, particularly in “Clinical outcomes in all patients” (line 184–185) and “Clinical outcomes in non-chordoma patients” (line 232) and the first paragraph of the “Discussion” section (line 266–267).

It would be desirable that a group-match study be performed to verify the real effectiveness of C-ion Rt versus RT. This comparison would help to understand the effectiveness of C-ion RT.

Response: We thank you for the insightful comment. We agree that group matching analysis of C-ion RT and X-ray RT would be useful to verify the effectiveness of C-ion RT. However, data of clinical outcomes in radical X-ray RT for bone sarcoma in our institution is both old and limited. Therefore, direct comparison of X-ray RT and C-ion RT is difficult and could not be performed in this analysis.

Authors should explain why they have decided to divide the entire population into two groups based on histology.

Response: We thank you for the insightful comment. The malignancy grade of chordoma and non-chordoma is different, and as a result, the clinical results are significantly different. Therefore, chordoma and non-chordoma were analyzed individually to evaluate the impact of C-ion RT in both groups. We added the following sentence in the statistical analysis section of “Materials and Methods”; “We analyzed clinical outcomes separately for all patients, chordoma patients, and non-chordoma patients because the malignancy grade of chordoma and non-chordoma is different.” (line 134–136).

Is there any difference in terms of OS, LC, and PFS between the group of patients who received previous treatment (chemo, surgery) versus the naïve group of patients who received C-ion RT?

Response: We thank you for the insightful comment. We did not observe any significant difference in OS, LC, and PFS between the patients who received prior treatment of chemotherapy and/or surgery and the treatment naïve patients. The 3-year OS, LC, and PFS of the patients who received prior treatment of chemotherapy and/or surgery versus treatment naïve patients were 80.2% versus 78.7%, 90.6% versus 83.3%, and 75.0% versus 44.4%, respectively (P = 0.640, 0.664, and 0.295, respectively). The corresponding outcomes for chordoma patients were 89.9% versus 100%, 90.9% versus 100%, and 81.0% versus 66.7%, respectively (P= 0.554, 0.435, and 0.941, respectively), and for non-chordoma patients, they were 59.8% versus 60.0%, 90.0% versus 71.4%, and 61.5% versus 0%, respectively (P = 0.713, 0.422, and 0.662, respectively). We added the OS and LC results for chordoma and non-chordoma patients in “Table 2”.

Round 2

Reviewer 1 Report

All comments have been adequately considered.

Author Response

Comment: All comments have been adequately considered.

Response: We sincerely thank you for evaluating our revised manuscript.

Reviewer 2 Report

The authors' response to my comment on IRB should be included in the manuscript.

Author Response

We sincerely thank you for evaluating our revised manuscript and for the important comments.

Comment: The authors' response to my comment on IRB should be included in the manuscript.

Response: We thank you for the important comment. We corrected the sentences as follows in the “Materials and Methods” section (line 86–90). “This retrospective study procedure was reviewed and approved by the Institutional Review Board and was used only medical records. The Institutional Review Board exempted our study from an individual informed consent, and the study was approved with an opt-out of notification regarding this analysis prior to this study.”

Reviewer 3 Report

The Authors have correctly and largely answered to all the Reviewers's questions and suggestions.

Author Response

Comment: The Authors have correctly and largely answered to all the Reviewer’s questions and suggestions.

Response: We sincerely thank you for evaluating our revised manuscript.
